# Environmental Regulation and China's Regional Innovation Output—Empirical Research Based on Spatial Durbin Model

**Yun Li** [1,2]**, Yingkai Tang** [1,3]**, Kun Wang** [1,3,]*****  **and Qiwei Zhao** [1,3]

[1]   Institute of Finance, Sichuan University, Chengdu 610065, China; liyuninscu@stu.scu.edu.cn (Y.L.); tang@scu.edu.cn (Y.T.); Zhaoqiwei@scu.edu.cn (Q.Z.)
[2]   Chengdu Municipal Xindu District People's Government of Sichuan Province, Chengdu 510100, China
[3]   Business School, Sichuan University, Chengdu 610065, China
*   Correspondence: liam_wang@stu.scu.edu.cn

**Abstract:** The sustainable development and innovation-driven development system has always been a guiding ideology for the Chinese government. Therefore, research on China's environmental regulation and regional innovation output is of great significance. Based on the provincial data of China from 2006 to 2016, this study uses many spatial econometric methods with the spatial Durbin model. We empirically analyze the relationship between environmental regulation and regional innovation output. The results show that (1) China's regional innovation output has significant spatial cluster and differentiation. Concerning the whole country, environmental regulation has a significant negative effect on regional innovation output, but its own spatial cluster phenomenon is not significant, and there is no space spillover. (2) There are differences between environmental regulation and regional innovation output in the eastern, central and western regions, in which the negative correlation between environmental regulation in the eastern region on regional innovation output has spillover effects in the region, and the direct effect in the central region is not significant, while the results in the western region are not significantly different from the full sample results. Finally, based on the research conclusions, we apply some policy recommendations from the perspectives of diversity of environmental policy, corporate innovation incentives, government officials' assessment, local government policy autonomy and sustainable development concept.

**Keywords:** environmental regulation; regional innovation spillover; regulatory overflow; spatial Durbin model; sustainable development

## 1. Introduction

Since the 1980s, the globalization and regionalization of the economy has been continued. In addition, the impact of purely violent capital accumulation on economic development is constantly being weakened. Improving the efficiency of capital utilization and completing the innovation-driven economy transition have already become issues of economic development. After the "18th National Congress of the Communist Party of China", Chinese President Xi has mentioned the concept of "new normal" on various occasions. Sustainable development has become one of the Chinese governments' core environmental and economic policies. The Chinese government has clearly stated that the Chinese economy should shift from factor-driven and investment-driven to innovation-driven. The realization of the innovation-driven intensive economic growth mode, the construction of an innovation system, the improvement of the national innovation system, the elimination of China's high-input and low-output production mode, and the promotion of innovation power are China's current major issues.

Although China's economic aggregate has been ranked second in the world, the traditional model of China's industrial economy has not been fundamentally resolved by the promotion of the "new normal" concept. According to estimates by Han Chao and Hu Haoran (2015), the traditional economic model of high energy consumption and high pollution accounts for 8% to 15% of the annual GDP loss [1]. Chinese government, enterprises, and people are affected by environmental problems such as smog, sandstorms, and soil erosion. Due to the Chinese government's increasing emphasis on the construction of ecological civilization, environmental regulation has naturally become an important factor affecting China's economic development. Thus, exploring the impact of environmental regulation on regional innovation output has practical significance for studying China's economic development.

We use the panel data of 30 provincial administrations (excluding Tibet Autonomous Region, Hong Kong, Macao, and Taiwan) from 2006 to 2016, in China, and construct environmental regulation indicators using three industrial waste emission datasets and investigate the relationship between environmental regulation and regional innovation output. We find that there is a very significant spatial autocorrelation between regional innovation output and environmental regulation. This study constructs a spatial Durbin model to measure the direct and spillover effects of environmental regulation on regional innovation output. The empirical results show that there are obvious strong and weak clusters and differentiation in China's regional innovation output. Environmental regulation has a significant inhibitory effect on regional innovation output, but the spatial cluster phenomenon is not obvious, and space spillover does not exist. Considering the specific situation of China, this study investigates the environmental regulations and regional innovation output in the eastern, western, and central regions of China respectively. The results show that there are differences between China's eastern, central, and western regions' environmental regulation and regional innovation output. The negative correlation of environmental regulation has spillover effects in the eastern region, and the direct effect of the central region is not significant, while the results of the western region are not substantially different from the full sample.

The innovations and academic marginal contributions of this study are, first, compared to previous literature, this study focuses on the spatial relationship between environmental regulation and regional innovation output. It focuses on whether environmental regulations in the region have an impact on the innovation output of the region and adjacent regions. Second, most of the literature is based on the perspective of industry or enterprise to study the relationship of environmental regulation and innovation, which ignores other economies, including non-profit economic groups. This study is based on regional macro perspectives and provides some reference for the formulation of local government environmental regulation policies.

In the next section, we review relevant prior studies and develop our research idea. Section 3 explains the research model and the selection process for the sample used in this study. Section 4 and Section 5 report the results of our empirical analyses and Section 6 concludes this study.

## 2. Literature Review

Prior research suggests that environmental regulation improves the innovation efficiency of high credit enterprises, but the positive impact cannot offset the total factor productivity loss (Popp and Newell, 2012), [2] which will reduce regional innovation output. Jorgenson and Wilcoxen (1990) prove that environmental regulation has increased the production costs of enterprises, occupied R & D investment, and thus inhibited the innovation output of the whole society [3]. Rubashkina, Galeotti, and Verdolini (2015) [4]; Zhu et al. (2019) [5] also state the negative effects of environmental regulation lead to the increased production costs based on different research perspectives. Cole and Elliott (2003), Levinson and Taylor (2010), and Cole, Elliott & Okubo (2010) further based on Japanese research and other literature also state that strict environmental regulations increase the economic burden of enterprises and inhibit their innovation power, which has led to a decline in the overall competitiveness of the country. In Reference [6–8], based on a sample of listed companies in China's A-share market, Zhou et al. (2019) state that the impact of environmental regulation on innovation output is negative.

However, some scholars such as Porter and Linde (1995), Li Xiaoping, Lu Xianxiang, Tao Xiaoqin (2012), Zhang Qian (2019), and others dismiss this view from an "innovative compensation" perspective [9]. They state appropriate environmental regulations can improve the enterprise's innovation output and regional innovation output. In Reference [10–12], Lanoie et al. (2011) argue that strict environmental regulations are conducive to promoting the use of new energy sources and saving more energy. Cost savings are conducive to the improvement of innovation output. This view is consistent with some Chinese scholars who state the relationship between China's environmental regulation and innovation output is J-type, U-s type, inverted N-type. (Tong Jian, Liu Wei and Xue Jing, 2016; Wang Yuguo, 2019; Shi Huaping, Yi Minli, 2019, etc.) [12–14]. Because the profit-oriented enterprise does not spend most production investment on environmental protection expenditures (Bu et al., 2013) [15], and there are huge uncertainties and risks in large-scale innovation investment (Zhao et. Al, 2015) [16], a sensible company tends to expand production in less risky ways to deal with environmental regulations. The innovation investment of the enterprise may not reach the threshold assumed in the mentioned literature. In summary, these studies are mainly based on the perspective of enterprises or regional industries. Considering the whole regional innovation output, this phenomenon disappears in measuring the overall regional innovation level, due to the horizontal sum of the innovation output levels of various enterprises or regional industries (the following empirical analysis will verify this).

According to the relevant researches for mechanism, we can find theoretical support from the direct effect and indirect effect perspectives. The direct impact of environmental regulation on regional innovation output can be divided into two aspects, first is environmental cost effect, another is the fittest survival (Gao Wei, Cheng Jinhua and Zhang Jun, 2018) [17]. From the environmental cost perspective, when the regional environmental regulation policy is issued and the production level of each economic entity remains unchanged, its pollution control cost and rent-seeking cost will inevitably increase, leading to a decline of the R & D investment. Secondly, potential entrants also consider the environmental risk and cost expenditure when measuring the entering cost, which reduces the innovation enthusiasm of enterprises. Thirdly, when the environmental cost of enterprises increases, enterprises will expand production scale or increase output, rather than choose innovation to pursue economic compensation (Chu Tingting, 2019) [18]. From the fittest survival perspective, as the environmental regulation cost increases, the operating cost of small private enterprises increases, and the survival ability decreases. In addition, the property right structure in this region deteriorates, which reduces the regional innovation output (Wang kun, Ji Xuanming and Xu He, 2018) [19].

The indirect impact of environmental regulations on regional innovation output is mainly based on "following cost hypothesis", which states that environmental regulations will increase the enterprises cost and decrease R & D investment, thus hindering enterprise innovation (Gray and Shadbegian,1998) [20]. Simpson and Lii (1996) argue that under static conditions, enterprises have made profit maximization, and the improvement of environmental regulation intensity in their regions will decrease their profits and reduce their innovation output capacity, especially small and medium-sized enterprises with insufficient funds [21]. The impact is more significant in China which is characterized by public ownership as the main part, and considerable development of the private economy. The analysis of the Iranian state-owned enterprises by Tajeddini and Trueman (2016) also gives a good explanation [22].

Due to the central government of China paying attention to the sustainable development strategy and the environmental problems caused by serious pollution problems such as PM2.5, whether the environmental problems can be solved has become part of the performance appraisal and promotion of the local officials. Especially in recent years, environmental issues and poverty alleviation became the focus of local officials (Lin, 2019) [23]. As the central government's requirements for environmental regulation become more stringent, the government invests a large amount of fiscal revenue in environmental governance, occupying the funds for R & D, and suppressing regional innovation output.

Prior studies on the impact of environmental regulation on innovation output are mainly from the micro perspective of enterprises or the medium perspective of industries. Relatively, studies on the overall impact of China's environmental regulation policies on regional innovation output are few. In addition, prior researches are seldom from a spatial perspective, ignoring the direct effect and spillover effect of environmental regulation. The first law of geography clearly states that everything is related to everything else, but near things are more related than distant things [24]. Additionally, Fredrisksson and Millimet (2002), Woods (2006) confirmed the spatiality of environmental regulation [25]. Thus, this study examines the impact of China's environmental regulation intensity on regional innovation output from the spatial perspective. Figure 1 shows the environmental regulation process.

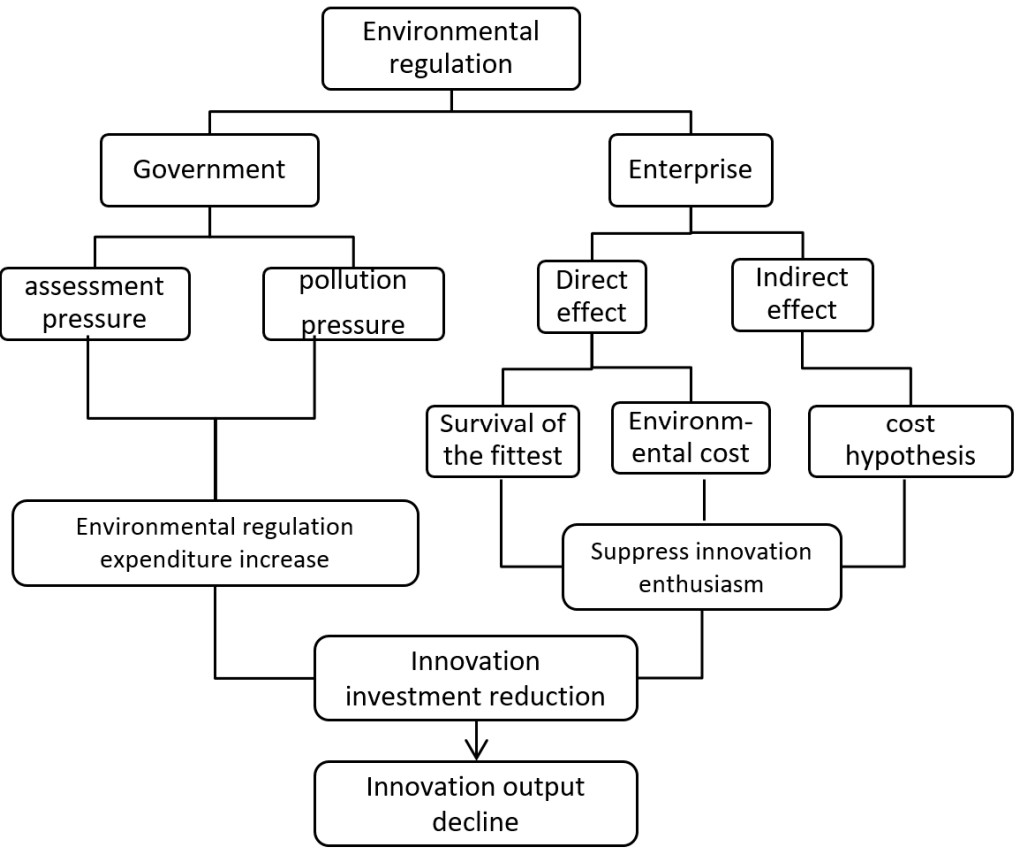

**Figure 1.** Environmental regulation process.

## 3. Material and Methods

### 3.1. Theoretical Model

According to the endogenous growth theory, scientific technological activities and progress can be regarded as endogenous production factors. Referring to the research of Zhang Chengdeng (2011), we consider $K_A$ and $L_A$ as the capital and labor of innovation investment, and $K_P$ and $L_P$ as capital and labor of production inputs [26]. Assuming that technological innovation is Hicks neutral, the production function can be written as $Y = A(K_A, L_A)F(K_P, L_P)$. According to Selden and Song (1995) [27], Shi Huaping, Yi Minli (2019) [14], etc., we consider the pollution control production as $G = \alpha A(K_A, L_A)F(K_P, L_P)$, and $0 < \alpha < 1$, $\alpha$ is the proportion of pollution input to total investment and also the response degree of region to environmental regulation I. In Reference [15,28] the regional pollution function $W = (Y, G)$ is determined by the total production Y and the pollution control production G, and $W'Y > 0$, $W'G < 0$. The Lagrangian function of regional net output maximization is:

$$\text{MAX} \prod = P[A(K_A, L_A)F(K_P, L_P) - \alpha A(K_A, L_A)F(K_P, L_P)] \tag{1}$$

$$\text{s.t. } W[A(K_A, L_A)F(K_P, L_P) - \alpha A(K_A, L_A)F(K_P, L_P)] = R \tag{2}$$

$$\text{FOC}: P(1-\alpha)A'(K_A, L_A)F(K_P, L_P) + \lambda \frac{\partial W}{\partial K_A} = 0 \tag{3}$$

$$-PA(K_A, L_A)F(K_P, L_P) + \lambda \frac{\partial W}{\partial \alpha} = 0 \tag{4}$$

$$\text{According to } (3)(4) \ \frac{\partial W}{\partial G} + \frac{\partial W}{\partial Y} = 0 \tag{5}$$

Model (5) shows the optimal condition of the environmental regulation constraint is that the marginal pollution increase of production is equal to the marginal pollution reduction of governance.

The technological innovation (I) is divided into production innovation (IA) and governance innovation (IG), and I = IA + IG, the technological innovation (I) is affected by production innovation (IA) and governance innovation (IG).

$$I'(A, G) = \frac{\partial I}{\partial W} \times \frac{\partial W}{\partial A} + \frac{\partial I}{\partial W} \times \frac{\partial W}{\partial G} \times \frac{\partial G}{\partial A} > 0 \tag{6}$$

$$\because \frac{\partial W}{\partial A} = \frac{\partial W}{\partial Y} \times F + \frac{\partial W}{\partial G} \times F \ \& \ \frac{\partial W}{\partial G} + \frac{\partial W}{\partial Y} = 0 \\ \therefore \frac{\partial W}{\partial A} = (1-\alpha)\frac{\partial W}{\partial Y} \times F \tag{7}$$

$$\therefore I'_A(A, G) = \left(\frac{\partial I_A}{\partial W} + \frac{\partial I_G}{\partial W}\right) \times (1-2\alpha)\frac{\partial W}{\partial Y} \times F > 0 \tag{8}$$

(I) $\frac{\partial W}{\partial K_A} = \frac{\partial W}{\partial Y} \times A' \times F + \frac{\partial W}{\partial G} \times A' \times \alpha \times F = (1-\alpha)\frac{\partial W}{\partial Y} \times A' \times F > 0$,
$\because W'_Y > 0, \therefore (1-\alpha)A' \times F > 0$. According to (3) $\lambda < 0$, According to (4) $\frac{\partial W}{\partial \alpha} < 0$.

Due to the proportion of governance investment in total investment increasing, the pollution level decreases.

(II) When $0 < \alpha < 1/2$, according to (8) $\frac{\partial I_A}{\partial W} + \frac{\partial I_G}{\partial W} > 0, \because \frac{\partial W}{\partial G} < 0$ and $\frac{\partial I_G}{\partial G} > 0$.
$\therefore \frac{\partial I_G}{\partial W} < 0, \therefore \frac{\partial I_A}{\partial W} > 0$.

When the proportion of pollution control in the region to total investment is less than 1/2, the environmental regulations increase, the pollution levels gradually decrease, and the production innovations also decline.

(III) When $1/2 < \alpha < 1$, $\frac{\partial I_A}{\partial W} + \frac{\partial I_G}{\partial W} < 0$, and $\because \frac{\partial I_G}{\partial W} < 0$, we cannot measure the coefficient of $\frac{\partial I_A}{\partial W}$. ① When $\alpha \to 1/2$, the environmental regulation is relatively loose, similar to (II), $\frac{\partial I_G}{\partial W} < 0, \frac{\partial I_A}{\partial W} > 0$ Environmental regulation inhibits regional innovation output. ② When $\alpha \to 1$, the environmental regulation investment is equal to the total production input in the region. Production and regional innovation output of the region is stagnant, and environmental regulation inhibits regional innovation output. The main reason is that excessive governance investment occupies the investment in technological innovation and hinders the technological innovation.

(IV) Assuming $\beta \in (1/2, 1)$, ① when $\alpha \in (1/2, \beta)$, environmental regulation is reasonable. As the environmental policies gradually increase, the pollution levels continue to decline, while production innovation levels rise against the trend, $\frac{\partial I_G}{\partial W} < 0$ and $\frac{\partial I_A}{\partial W} < 0$, regional environmental regulation promotes technological innovation. ② When $\alpha \in (\beta, 1)$, environmental regulations are strict, $\frac{\partial I_G}{\partial W} < 0$ and $\frac{\partial I_A}{\partial W} > 0$, environmental regulations inhibit technological innovation.

According to (I–IV), the relationship between environmental regulation and regional innovation output is inverted N type, but the pollution input accounts for more than 1/2 of the total investment ($1/2 < \alpha < 1$), it may appear in a firm or an industry. However, it cannot appear in China, for it is impossible for a province investing more than 1/2 of total investment in pollution control.

According to the theoretical model, the core hypothesis of this study is, in China, the improvement of environmental regulation intensity will inhibit regional innovation output in a region. In addition, based on the basic principles of "the first law of geography", this study assumes that the impact of environmental regulation policy is spatial.

### 3.2. Model Building

In order to examine the impact of environmental regulation on regional innovation output, based on the theoretical model above and referring to the studies of Yang et al. (2019) and Liu et al. (2019) [29,30], we establish the following spatial Durbin model (SDM):

$$RCP_{it} = \varrho_0 W_{i,j} + \beta_1 ERS_{i,t-1} + \beta_2 W_{i,j} ERS_{i,t-1} + \beta_3 X_{i,t} + \beta_2 W_{i,j} X_{i,t} + \mu_i + \lambda_t + \varepsilon_{i,t} \tag{9}$$

In which $P_{i,t}$ is the innovation output of the province $i$ in the year $t$, and $ERS_{i,t}$ represents environmental regulation. Because environmental regulation has a lagging effect on regional innovation output, we lag $ERS_{i,t}$ in the regression. $X_{i,t}$ represents all the control variables in this study, $W_{i,j}$ is the spatial weight matrix, $\mu_i$, $\lambda_t$ and $\varepsilon_{i,t}$ are spatial effects, time effects, and random disturbance terms, $\rho_0$ is the spatial lag regression coefficient; $\beta_i$ is the regression coefficient.

The spatial weight matrix $W_{i,j}$, which reflect the specific spatial relationship between regions, is an indispensable part of the spatial model. The spatial matrix mainly includes first-order adjacent spatial weight matrix, geographic distance spatial weight matrix, and economic distance spatial weight matrix. In China, provincial-level research usually uses the neighborhood criterion to adopt the adjacent spatial weight matrix (Hu and Zhao) [31].

The first-order adjacency space weight matrix is:

$$W_{ij} = \begin{cases} 1, & \text{Province } i \text{ is adjacent to } j \\ 0, & \text{Province } i \text{ is not adjacent to } j \end{cases} \tag{10}$$

If two provinces are post-adjacent, $W_{i,j}$ equals 1, and 0 otherwise. Due to China's particular geographical location, although Hainan Province and Guangdong Province and Guangxi Zhuang Autonomous Region are across the sea, they are still adjacent to each other set equal to 1. Because communication has regional dependence, and the shorter the distance, the lower the cost, the communication between adjacent provinces is closer. In addition, policy-oriented cross-provincial cooperation does not affect neighboring provinces' communication. For example, the GDP between Guangxi and Liaoning is not much different. Their geographic distance is very far, but the economic distance is very close. However, because the two provinces are located in the southwest and northeastern regions of China, the environmental characteristics and environmental regulation levels are inevitably different. Some literatures calculate three matrices separately and select the best results for analysis, which is easy to fall into the "econometric trap". So we perform row normalization on the "0–1" spatial weight matrix $W_{i,j}$.

### 3.3. Variables Measurement and Data Sources

Regional Innovation Output (RCP): This study uses the popular patent grants (New practical patent) as a surrogate indicator for regional innovation output to comprehensively reflect the actual number of technological innovations. Taking into account the impact of the size and population base of different provinces in China on the number of patent grants, this paper finally decided to use the number of patent grants per capita (unit: item/million) to measure regional innovation.

Environmental Regulation (ERS): The idea of constructing environmental regulation indicators in this paper is to study the city by constructing the relative positions of different pollutant emission intensities across China and then weighting the relative levels of pollution intensity of the average city. For, the extent of efforts in environmental pollution control, this article draws on the practice of Wang Guoyu (2019) [12]. The specific steps are as follows: (I) Calculate the environmental pollution

emission intensity of the province $i$: $E_{v,it} = \frac{e_{v,it}}{Y_{it}}$, in which $e_{v,it}$ represents the total amount of pollutants $v$ of the province $i$ in the year $t$; $Y_{it}$ represents the actual industrial output value of the province $i$ in the year $t$ (In 2003, $Y_{it} = 100$); $E_{v,it}$ is the emission intensity of the pollution $v$ of the province $i$ in the year $t$. (II) Calculate the national environmental pollution emission intensity $\hat{E}_{v,it} = \sum_{i=1}^{30} \frac{e_{v,it}}{Y_{it}}$, in which $\hat{E}_{v,it}$ is the emission intensity of pollution $v$ of China in the year $t$. The pollution emission includes industrial wastewater emission, industrial $SO_2$ emissions, and industrial soot emissions. (III) Calculate the relative intensity of environmental pollution emissions $ER_{v,it} = E_{v,it}/\hat{E}_{v,it}$, where $ER_{v,it}$ is the Relative position of the emission intensity of the pollutant $v$ in the province $i$ in the year $t$. The larger $ER_{v,it}$ is, the more the emission intensity of the pollutant $v$ in the province $i$ in the year $t$ is, which if it is relatively high across the country, indicates that the environmental control intensity is looser. (IV) Calculate a comprehensive index of local government environmental controls. Because $ER_{v,it}$ is a dimensionless variable, so we can get $ER_{it} = 1/3(E_{1,it} + E_{2,it} + E_{3,it})$. (V) In order to be consistent with the expected coefficient of the theoretical hypothesis, we inversely processed the index $ERS = 1/ER_{it}$. The higher the pollution emission comprehensive index is, the higher the government's environmental pollution control is, and the stricter the environmental standards are. In the contrary case, the environmental control is weaker. Due to the lack of some cities' data, when calculating the provincial data, we use the relationship between the industrial output value of the city and the industrial output value of the provincial capital cities to estimate.

Considering the driving factors related to regional innovation output, the control variables are foreign direct investment, regional infrastructure construction level, regional human capital level, regional fixed asset investment level, and government governance.

Foreign Direct Investment (FDI): Since the reform and opening, China's regional innovation capability has been greatly improved. Grossman and Helpman (1991) argue that trade openness can promote innovation by bringing about competitive effects, diffusion of technology and innovative ideas. As per Reference [31], as an important factor affecting innovation, regional openness, should be used as a control variable to examine regional innovation performance. Therefore, the foreign direct investment is included as a control variable. (Convert the average exchange rate into RMB at the end of the year).

Regional Infrastructure Construction Level (RIC): Differences in regional infrastructure construction levels can lead to differences in regional innovation performance. Improvements in infrastructure such as regional communications can increase regional innovation levels. The various communication tools in regional communication is increasingly important, and communication requires the region to provide the necessary long-distance optical cable facilities. Following Reference [32], thus, we use the length of regional long-distance optical cable lines to measure the regional infrastructure construction level.

Regional Human Capital Level (RHC): Following Reference [33], We calculate the average level of education according to the current Chinese education year system. The specific formula is: average education level = (college and above population * 16 + high school population * 12 + junior high school population * 9 + primary school population * 6)/population over 6 years old. We assume that the average age of education for college graduates and above is 16 years. Due to the lack of this data in 2010, we use the growth rate of relevant data to estimate.

Regional Fixed Assets Investment Level (RAI): The ratio of total fixed assets investment to total population in all provinces.

Government Governance (GOV), measured by the rate of non-nationalization: Non-state-owned rate = 1—the total industrial output value of Chinese state-owned enterprises/the industrial output value of enterprises above scale.

Except for environmental regulation (ERS), we use annual provincial panel data of China from 2006 to 2016. Due to the lag of variables involved in environmental regulation, the data range is from 2005 to 2016. Considering the comprehensiveness and availability of data, the provincial administrative regions do not involve Hong Kong, Macao, Taiwan, or Tibet autonomous region. We collected the

data for this study from the Wind database and calculated the regression results using the Stata 15MP software package. We eliminated the influence of outliers by winsorizing all continuous variables at the 1% level.

## 4. Results and Discussion

### 4.1. Sample Description

Figure 2 shows the total amount of industrial wastewater, industrial SO$_2$, and industrial soot emissions from 30 provincial administrations in China from 2006 to 2016. Figure 1 shows that the "Industrial Wastes" emissions of the three provinces of Shanxi, Shandong, and Hebei are clearly ahead of other provinces. These provinces are traditional energy-consuming provinces and are also smoggy in recent years and are geographically connected. In areas with high levels of economic and technological innovation such as Beijing, Tianjin, Shanghai, Fujian, Guangdong, etc., their sewage emission is generally lower than the national average. In remote areas of western China such as Ningxia, Qinghai, Guizhou, and Yunnan, due to the backward economic development, their sewage emission is not high. In general, there is a spatial cluster in the overall emission volume and there is a regional gap. It is meaningful to study the subsamples of East, Central, and West China.

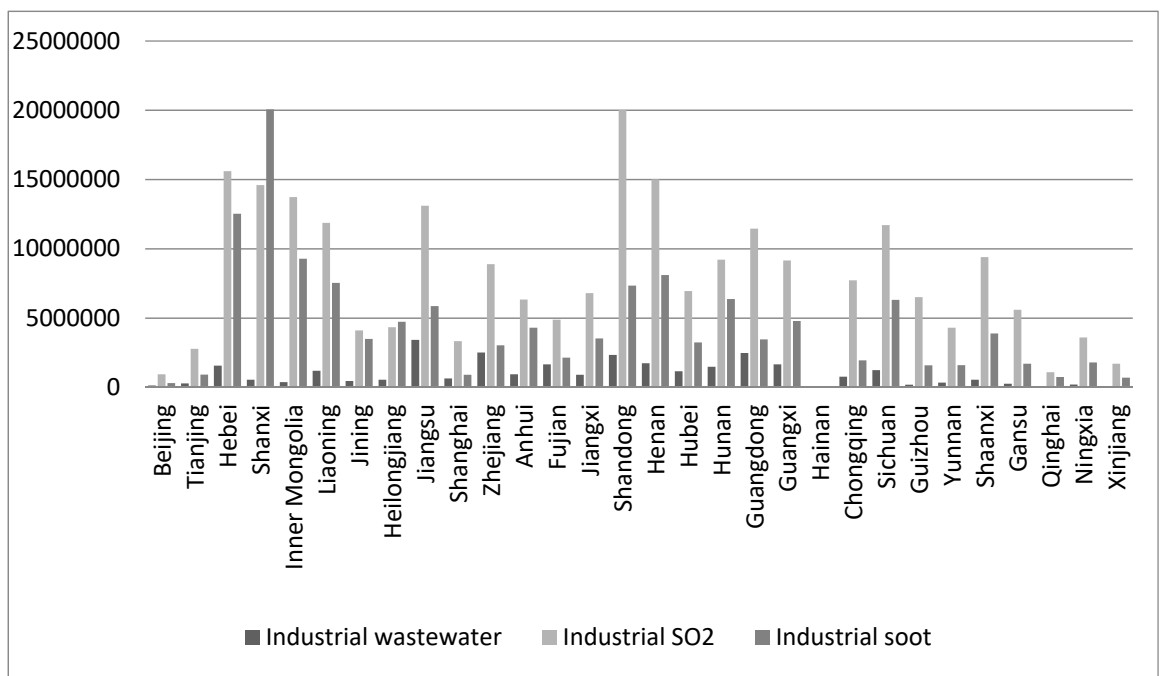

**Figure 2.** Total province pollutants emission in 2006–2016.

Figure 3 shows the average growth rate of innovation output in all provinces in China from 2006 to 2016, which reflect the change level in regional innovation output in each region over the past 11 years. It shows that China's innovation output has remained at a high level for more than a decade. With the strategic advancement of China's Hainan Free Trade Zone and the support of the central government, the second-ranked Hainan Province's innovation output growth rate is relatively fast. In economically developed areas such as Beijing, Tianjin, Shanghai, Zhejiang, and Jiangsu, the growth rate is relatively stable, maintaining around 15%–20%.

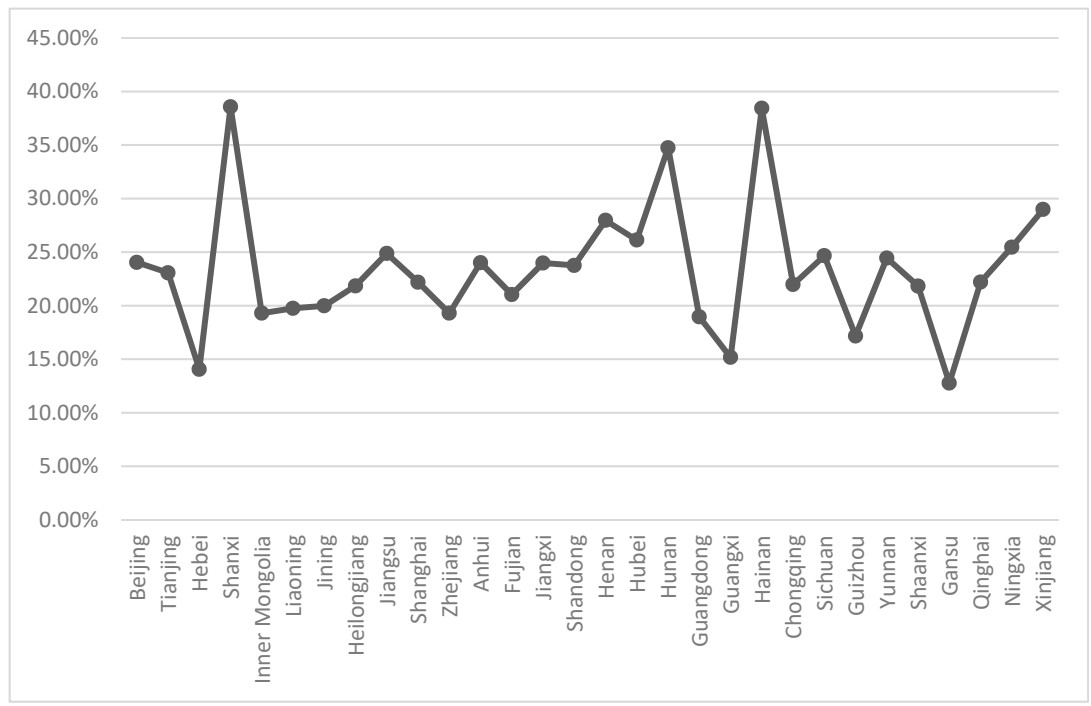

**Figure 3.** Average growth rate of innovation output in all provinces in China from 2006 to 2016.

*4.2. Spatial Correlation Test*

The global spatial autocorrelation test mainly investigates the spatial cluster of the entire spatial sequence. The most popular test method is the Moran's I test. We calculated the Moran's I index by using the first-order adjacent spatial weight matrix of 30 provinces to test the global spatial autocorrelation of regional innovation output (RCP) and environmental regulation (ERS). Table 1 shows the relevant statistical results.

**Table 1.** Moran's I test results of RCP and ERS.

| Year | RCP | | ERS | |
|------|-----------|---------|-----------|---------|
| | Moran's I | *p*-Value | Moran's I | *p*-Value |
| 2006 | 0.263 | 0.004 | 2.855 | 0.002 |
| 2007 | 0.273 | 0.005 | 1.955 | 0.025 |
| 2008 | 0.286 | 0.006 | 2.041 | 0.021 |
| 2009 | 0.264 | 0.023 | 2.331 | 0.010 |
| 2010 | 0.251 | 0.031 | 2.255 | 0.012 |
| 2011 | 0.183 | 0.027 | 2.642 | 0.004 |
| 2012 | 0.187 | 0.014 | 2.550 | 0.005 |
| 2013 | 0.194 | 0.024 | 2.517 | 0.006 |
| 2014 | 0.195 | 0.033 | 2.334 | 0.010 |
| 2015 | 0.200 | 0.034 | 2.590 | 0.005 |
| 2016 | 0.194 | 0.038 | 2.453 | 0.007 |

It shows that the Moran's I index of China's regional innovation output and environmental regulation in 2006–2016 is positive, indicating that there is positive spatial autocorrelation. Additionally, the data of all years passed 5% significance test, rejecting the null hypothesis that "there is no spatial autocorrelation", indicating that there are significant positive spatial autocorrelations in regional innovation output over the years, and the cluster phenomenon is more obvious.

We use Moran's I scatter plot to examine the degree of association and correlation between individuals in the space. The Moran's I scatter plot is based on a normalized Cartesian coordinate

system in which the abscissa is the attribute value for each region, and the ordinate is the average values of the neighboring regions. The Moran's I scatter plot consists of four quadrants. The first quadrant (HH) represents the aggregation of high and high values. The set of regions formed by the region within the quadrant and its surrounding regions is called "Hot spot"; the second quadrant (LH) indicates that the low value and the high value are concentrated, and the sample value of the region is lower, and that of the surrounding region is higher; the third quadrant (LL) is completely opposite to the first quadrant (HH), indicating that the low value and the low value are concentrated. The set of regions formed in the region within the quadrant and its surrounding region is called "Cold spot"; the fourth quadrant (HL) is similar to the second quadrant (LH), which reflects that the sample value of the region is higher and the sample mean of the surrounding region is lower.

Based on the starting year of each "five-year plan" in China, we draw the Moran's I scatter plots of China's regional innovation output in 2006, 2011, and 2016, as shown in Figures 4–6.

Figures 3–5 show the spatial cluster distribution of China's regional innovation output has not changed significantly in the past 11 years. Most provinces are in the third quadrant (LL), and only about 20% of the provinces are in the first quadrant (HH), indicating that the proportion of strong and weak innovative provinces in China is stable, and the innovation output of each region cannot be regarded as an independent observation. It is insignificant to explore the Moran scatter plot of explanatory variables, so we are no longer reporting the Moran scatter plot of environmental regulation.

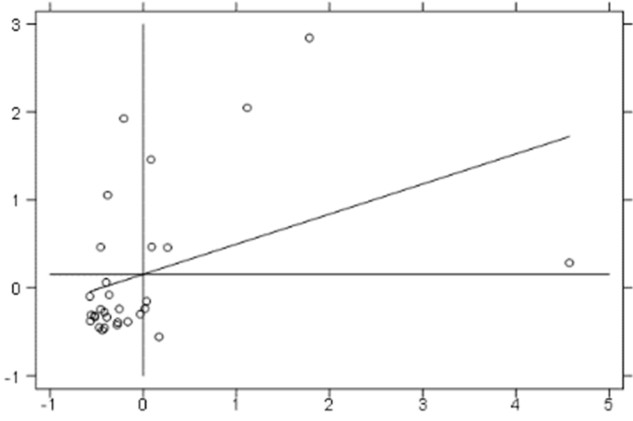

**Figure 4.** 2006 Moran scatter plot.

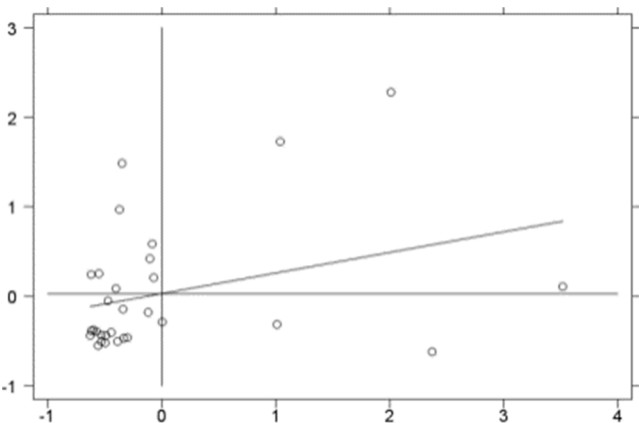

**Figure 5.** 2011 Moran scatter plot.

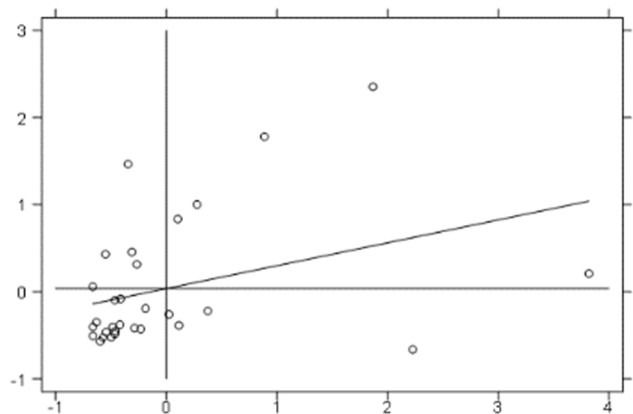

**Figure 6.** 2016 Moran scatter plot.

## 5. Spatial Econometrics Analysis

### 5.1. Model Selection

We use the Lagrangian multiplier test (Breusch and Pagan, 1980) to determine whether to adopt the mixed regression model or the random effects model, [34] the value of the $\chi^2$ statistic is 529.03, and the corresponding *p* value is 0.0000. Therefore, the above results reject the null hypothesis that "mixed regression is feasible", so we use the random effect model instead of the mixed regression model. The heteroscedasticity of panel data may lead to the traditional Hausman test fails, so we use the heterodyne-stable Hausman test (Durbin–Wu–Hausman test) to determine whether to adopt the fixed effect or the random effect. The Sargan–Hansen statistic is 85.179 and the corresponding *p* value is 0.0000. Thus, we adopt the fixed effect model (Dunbar, Li and Shi, 2016) [35]. Finally, considering the existence of spatial fixed effects, time-period fixed effects, or spatial and time-period fixed effects, we use the F test. The value of the spatial statistic and time effect F statistic is 411.6 and 3.871. At the 1% significance level, the time effect is not significant. Therefore, we use the spatially fixed spatial Durbin model.

Although many literatures simplify the spatial Durbin model, it can also be easily implemented in Stata software using commands such as "spregdpd". However, SDM is the most general form. We can use regression results to determine whether it can be simplified, so this study will not report this content.

### 5.2. Estimated Results of the Spatial Econometric Model

We calculate the model (9) based on the 30 provincial panel data (excluding Hong Kong, Macao, Taiwan, and Tibet regions) from 2006 to 2017. Maximum Likelihood Estimation (MLE) is one of the mainstream estimation methods for spatial econometric models. It can alleviate the effects of endogeneity. There is no lag term of the explanatory variables in the regression model. MLE is more efficient than GMM (LeSage and Pace, 2009) [36]. Since the two-way fixed effect needs to be reported, the modified MLE method is finally used for estimation. In order to prove that the space Durbin model of spatial fixed effect is more efficient and report the results of other regression methods, Table 2 shows the empirical results.

**Table 2.** Estimated Results of the Spatial Durbin Model.

| Independent Variable | Spatial Durbin Model | | | Ordinary Panel Model | |
|---|---|---|---|---|---|
| | Spatial Fixed Effects | Time-Period Fixed Effects | S and T Fixed Effects | Individual Fixation Effect | Two-Way Fixed Effect |
| ERS | −1.718 *** (−6.77) | −1.4592 *** (−6.40) | −1.6944 *** (−6.47) | −1.237 *** (−3.50) | −2.3203 *** (−9.53) |
| FDI | −0.000385 ** (−2.38) | −0.0004061 *** (−3.37) | −0.000401 ** (−2.32) | −0.000448 *** (−5.41) | −0.00049 ** (−2.94) |
| RIC | 0.0006283 *** (8.02) | 0.000547 *** (7.34) | 0.0006581 *** (8.03) | 0.00038 *** (3.30) | 0.000617 *** (6.53) |
| RHC | 3.4012 ** (1.97) | 4.58904 *** (2.83) | 3.03736 * (1.81) | 4.7478 *** (3.65) | 0.3598 (0.16) |
| RAI | 0.61343 ** (2.27) | 0.63757 ** (2.28) | 0.5727 ** (1.96) | 1.9169 *** (3.51) | 0.6163 * (1.82) |
| GOV | 12.95564 ** (2.13) | 9.9774 * (0.091) | 13.6025 ** (2.22) | 3.66134 (0.93) | 9.9445 * (1.84) |
| W*RCP | 0.40925 *** (2.04) | 0.42162 *** (7.00) | 0.35677 *** (5.49) | — | — |
| W*ERS | 0.6633 (0.72) | 0.84613 (1.02) | 0.42032 (0.44) | — | — |
| W*FDI | 0.0007129 *** (4.000) | 0.000840 *** (6.42) | 0.00068 *** (3.28) | — | — |
| W*RIC | 0.0002167 (1.52) | 0.000184 (1.37) | 0.000361 ** (2.14) | — | — |
| W*RHC | −3.4636 * (−1.73) | −3.6776 * (−1.77) | −5.612 * (−1.74) | — | — |
| W*RAI | −0.3369 (−0.60) | −0.5167 (−0.93) | −0.3921 (−0.54) | — | — |
| W*GOV | −24.7409 * (−2.05) | −22.1116 ** (−1.99) | −21.548 * (−2.04) | — | — |
| Constant | — | — | — | −50.4153 *** (−5.47) | −14.329 (−0.76) |
| Log-L | −1004.4939 | −1081.8313 | −1010.9708 | — | |
| R2 | 0.5821 | 0.5721 | 0.5651 | 0.3152 | 0.5612 |

***, **, and * indicate statistical significance at the 1%, 5%, and 10% levels, respectively. Z statistics in parentheses.

Table 2 shows with the good fit R2 that the spatial Durbin model is superior to the ordinary panel model. Among the effects of the spatial Durbin model, the R2 fitting value and the Log-L likelihood value of the spatial fixed effect model are also superior to the bidirectional fixed effect and the time fixed effect. We validate the space Durbin model of spatial double-effect fixed effect from the empirical data is efficient and persuasive. However, there is no substantial difference in the regression results of the three models, and the regression methods have little effect on the empirical results. The regression results in Table 2 show that, the coefficient of *W\*RCP* is 0.40925, and it is significant at the 1% level, indicating that regional innovation has the significant positive spatial cluster, the development of innovation in a region will drive the innovative development of neighboring provinces, which reflects the spatial distribution characteristics of regional high-level cluster and low-low concentration. Although the coefficient of *W\*ERS* is positive, it is not significant, and spatial cluster effect of environmental regulation does not exist across the country. (The author also tries to introduce the square term of ERS and the square and third terms of ERS, and find the core explanatory variables cannot be significant at the same time in both cases. So U-type and N-type relationship does not exist, which is consistent with the theoretical analysis above.)

Elhorst (2014) shows that the explanatory variables of the spatial Durbin model and the coefficients of the spatial lag explanatory variables are not explanatory, and it is meaningless to discuss the saliency and numerical values of the coefficients. In Reference [37] he states that the model should be interpreted in direct and indirect effects. Thus, we decompose the regression coefficients of the spatial panel model. Table 3 shows the results.

**Table 3.** Direct and indirect effects of the spatial fixed effect space Durbin model.

| Direct Effects | Coefficient | t | *p* | Indirect Effects | Coefficient | t | *p* |
|---|---|---|---|---|---|---|---|
| ERS | −1.7142 | −5.87 | 0.000 | ERS | −0.07062 | −0.05 | 0.963 |
| FDI | −0.000329 | −2.04 | 0.042 | FDI | 0.000894 | 3.21 | 0.001 |
| RIC | 0.000675 | 8.38 | 0.000 | RIC | 0.000736 | 3.76 | 0.000 |
| RHC | 3.17485 | 1.97 | 0.049 | RHC | −3.4980 | −1.25 | 0.211 |
| RAI | 0.6362 | 2.36 | 0.018 | RAI | −0.06457 | −0.08 | 0.940 |
| GOV | 10.9385 | 1.85 | 0.064 | GOV | −30.762 | −1.67 | 0.095 |

Table 3 shows that the direct effect coefficient of ERS is −1.7142, and its *p* value is 0.000, which is significant, indicating that the region with more stringent environmental regulations has lower regional innovation output, which also supports the results above. The main reasons are (I) The increase in environmental regulation investment occupies the government and enterprises' innovation investment. Specifically, the government increases the intensity of environmental regulation in the jurisdiction, which increases the financial investment in environmental protection. With the same fiscal revenue, the government's support for innovation decreases. Considering enterprises, as the intensity of environmental regulation increases, the pollution costs and penalties increase, leading to the increasing of the investment in environmental protection funds. Due to the guaranteed basic production expenditures, innovation investment is more likely to be occupied on a large scale. (II) Due to the pressure on environmental regulation increases, the cost of the enterprise increases. Because the time lag and uncertainty of innovation activities, most companies tend to expand production scale rather than invest in uncertain R & D activities in order to ensure profits.

The *p* value of the indirect effect of ERS is 0.963, which is insignificant. The indirect benefits reflect the impact of the province's independent variables on the innovation output of neighboring provinces or the influence of independent variables of neighboring provinces on regional innovation output in the province. Because the economic development level of China's provinces is very different from the industrial structure and the difficulties between economic development and environmental protection are not the same, environmental regulation has no spillover effect on regional innovation output in China. For example, Fujian Province is China's first open city. Quanzhou has been the most important port in southern China since the Song Dynasty. It is also the most densely populated area of

China's light industry, especially the textile industry. Fujian is almost covered by mountains so the environmental regulation pressure is small. Jiangxi Province is a region with a relatively backward economy in China. Its environmental protection pressure is high, and the economic development pressure is high. When Jiangxi Province officials consider environmental regulation issues, they are not affected by Fujian's environmental regulation policies.

### 5.3. Regional Tests

Due to China's vast territory, the resource and economic development levels of the eastern, central, and western regions are very different. The national-level analysis does not fully explain the effect level at which environmental regulations affect the industrial structure of each region (Fang, Wang, 2016). This study refers to the standard of Yang et al. (2019) [29] to divide the 30 provincial-level administrative units into eastern, central, and western regions. Figure 7 shows the division.

Table 4 shows that the significance of W*RCP is good and the coefficients are positive, indicating that there is a positive cluster of innovative output in all three sub-regions. (I) Both the direct and indirect effects of the eastern region are significant, demonstrating the negative correlation between environmental regulation and regional innovation output in the region, ① due to the high level of financial and industrial cluster in the eastern region, frequent exchanges between provinces, more detailed labor division between provinces, and higher dependency. It is also the commonality of global economic development. The cost increase brought about by a province's environmental regulation policy will affect other provinces. For example, under the policy pressure of the sustainable development strategy, in order to decrease the PM2.5 index, Hebei Province tends to temporarily close some high-consumption raw material production enterprises in winter, which will increase the cost of other provincial partners. The central and western regions are characterized by agriculture and resource industries, and the impact will be even smaller. ② The eastern region has a good foundation for innovation and economy. However, the comparison and competition between regions are also fierce, due to the "neighbor imitation effect" and "demonstration effect". This kind of competition is mostly between economically developed regions. The policy formulation of provinces in economically developed regions is generally not affected by the central and western regions, but the economically developed provinces are the first to compete with each other. For example, the competition for the introduction of high-level talents among provinces appears firstly in the eastern regions of Zhejiang and Jiangsu, then the central and western regions propose the relevant policies, but the strength and appeal of the policy is far less than in the eastern region. (II) The direct effect of the central region is not significant, which is caused by the second phase of industrial development. The central region is China's major agricultural and basic industrial province. The environmental regulation is mainly responsible for the traditional industries such as steel and coal. Recently, the central region has developed many tertiary industries with the promotion of tax incentives and other measures. Environmental regulations have little impact on them. (III) The direct effect of the western region is significantly negative, and the indirect effect is not significant, indicating that the spillover effect does not exist in this area. The main reason is that the dependence on resources of the western provinces is more serious than on those of the central and eastern regions. Additionally, the resource reserves, the core industries, the pressures of environmental regulation, and the demand for innovative output of the provinces in western China are different, thus spillover effect does not exist, which is confirmed by Wang (2011) [38] and Zhao (2016) [39].

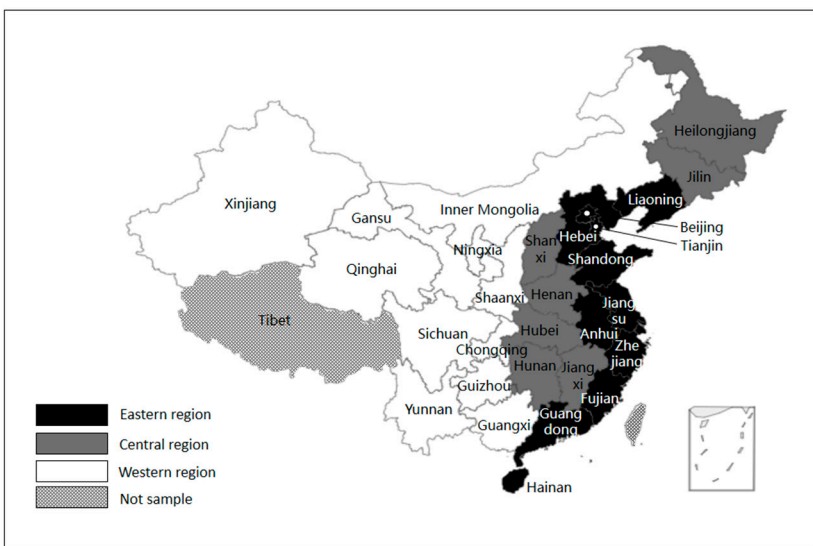

**Figure 7.** Sample area description.

Table 4 shows the correlation regression of main indicators (The significance level of the unreported part is not significantly different from the national sample, and the coefficient significance and value of SDM have no practical significance, so we will not report).

**Table 4.** Regression results of regional Tests.

| Regional | W*RCP | ERS | W*ERS | Direct Effects | Indirect Effects | R² | Log-L |
|---|---|---|---|---|---|---|---|
| Eastern Region | 0.1217 ** (1.15) | −1.6882 *** (−10.65) | −0.5192 (−1.47) | −1.7170 *** (−10.65) | −0.7906 ** (−2.36) | 0.8497 | −327.7573 |
| Central Region | 0.3690 *** (4.57) | −2.0886 (0.322) | 7.9967 *** (2.66) | −2.0419 (−0.60) | 6.2381 (1.06) | 0.6851 | −228.0180 |
| Western Region | 0.3248 *** (2.90) | −7.4365 *** (−3.15) | −2.8220 (−0.62) | −7.4687 *** (−2.79) | −0.6259 (−0.09) | 0.6256 | −369.666 |

***, **, and * indicate statistical significance at the 1%, 5%, and 10% levels, respectively. Z statistics are in parentheses. The three sub-regressions are all related to the study within the region and cannot exclude the interaction between the three regions of the eastern, central, and western regions. For example, the spatiality between Anhui Province in the eastern region and Henan Province in the central region cannot be excluded by empirical results. In addition, due to the subsample size, the statistical risk is also higher.

## 6. Conclusions

Based on the panel data of 30 provincial administrative units in China from 2006 to 2016, this study investigates the relationship between environmental regulation and regional innovation output using spatial economic measurement. First of all, we find that there is a significant spatial positive correlation between environmental regulation and regional innovation output. The strong and weak cluster of the past 11 years is stable, and there is a typical phenomenon of HH-LL differentiation. Secondly, the regression results show that regional innovation has the characteristics of significant positive spatial cluster, the development of innovation in a certain region will drive the innovation and development of neighboring provinces. There is no spatial cluster effect of environmental regulation across the country. Third, at the national level, environmental regulation has a significant negative effect on regional innovation output, but the spatial spillover does not exist. It means the intensity of environmental regulation in neighboring provinces has little impact on the province's innovation output across the country. Fourth, we divide China's provincial administrative units into eastern, middle, and western and find that there are differences between environmental regulation and regional innovation output, in which the negative correlation between environmental regulation in the eastern region on regional

innovation output has spillover effects in the region, and the direct effect in the central region is not significant, while the results in the western region are not substantially different from the full sample.

The main policy implications are: (I) Maintain the concept of sustainable development and find a balance between the environment and the economy. In 1997, the 15th National Congress of the Communist Party of China identified sustainable development as a strategy that must be implemented in modernization. Sustainable development has always been one of the guiding ideologies for China's economic development. It has been proved that this strategy has played an important role in China's environmental protection. Therefore, we should insist on the concept of sustainable development and keep a balance between the environment and the economy. (II) The government should rationally allocate various environmental regulations and abandon the "one size fits all" approach. The empirical results show that there is a negative correlation between environmental regulation and regional innovation output. However, when conducting environmental regulation, the government should formulate more detailed environmental regulation plans according to the specific conditions of different industries and even different enterprises and reduce the negative externalities of environmental regulation. (III) Increase the economic compensation system of innovation and reduce the innovation cost of enterprises. According to the "following cost" hypothesis, we find the environmental regulation indirectly increases the opportunity cost of enterprise R & D activities, and economic compensation for enterprises can reduce the opportunity cost of R & D activities in the region. (IV) Establish a comprehensive evaluation system for official assessment, which does not use GDP as a "hard currency" for evaluation nor uses the environmental protection as the only rule of officials for the "green mountains and green mountains". (V) The government should enhance the local government's policy autonomy and improve the coordination. According to the empirical research, the relationship between environmental regulation and regional innovation output varies in different provinces. National government should give the authority of policy formulation to local governments and play a role of supervision and coordination, which can improve the efficiency of environmental regulation and reduce the negative impact of environmental governance.

**Author Contributions:** Conceptualization, K.W.; data curation, Y.L. and Q.Z.; funding acquisition, Y.T.; investigation, Y.T.; software, K.W.; writing—original draft, Y.L.; writing—review and editing, K.W. and Q.Z.

**Funding:** This research was funded by the National Natural Science Foundation of China (No. 71072066), Sichuan University (No. SKGT201602, No. 2018HHF-42), and the Department of Science and Technology of Sichuan Province (No. 2018JY0594).

**Conflicts of Interest:** The authors declare that they have no conflicts of interest.

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
