# Peer review of "Environmental Regulation and China’s Regional Innovation Output—Empirical Research Based on Spatial Durbin Model"

_sustainability, doi:10.3390/su11205602_

Round 1

Reviewer 1 Report

attached

Author Response

Dear reviewer:

I am glad that you have carefully reviewed my paper and made valuable comments. I read your comments carefully and revised the paper. I explain your review comments as follows:

1, We rewrote the contribution of the article on your recommendation.

2, About endogeneity:

The estimator obtained by spatial Durbin model is unbiased, so it is almost unnecessary to discuss its endogenous problem. Some monographs explain this in detail. For example: LESAGE J, PACE R K. Introduction to Spatial Econometrics. Boca Raton: CRC Press, 2009.

The spatial Durbin model provides a common starting point for spatial metrology models, which is well suited for model selection and is a suitable framework model for capturing different types of spatial spillover effects (Le Sage and Pace, 2009). Elhorst (2010) calls it a milestone in the history of applied spatial metrology: first, it guarantees an unbiased estimation of model coefficients, whether the data is a spatial lag model or a spatial error model; second, it does not impose restrictions on the scale of potential spatial spillover effects.[Elhorst. Applied Spatial Econometrics: Raising the Bar.Spatial Economic Analysis,2010,5( 1) : 9 – 28]

3, We have added some new references, including some you recommended. And in this part, we have strengthened the discussion and explanation of regional innovation. We added a Roadmap (Fig. 1) .The references you recommended are excellent and helpful for us. We want to thank you again.

4, In abstract ,“negative”is coefficients of Direct Effects of Spatial Durbin Model(Table 3). In the conclusion , “positive” is spatial autocorrelation (Fig4-Fig6). Those are represented in the second and third points of the conclusion separately.

5, We checked the grammar and words of the article. If paper can be accepted, we want to choose the article polishing service.

According to the opinions of other reviewers, we have also made the following revises:

1, We supplemented the analytical steps of the theoretical model.

2, Strengthen the analysis of empirical results. It's mainly in “Estimated Results of the Spatial Econometric Model” and ” Regional Tests”.

We hope you satisfied with the revision of our article. If you think there are any shortcomings in our article, please point it out directly. We will revise it to your satisfaction.

Thank you again for your work.

Best wishes!

Authors of this paper.

Reviewer 2 Report

Thank you for inviting me to review the manuscript. Here are my comments:

More insightful arguments are required for regional innovation. More explanation is required to justify the theoretical model The research question should be justified The novelty of research should be discussed. The conclusions are weak and needs some more explainations.  The following papers might help you improve your work: Tajeddini, K., and Trueman, M.,. 2016. Environment-Strategy and Alignment in a Restricted, Transitional Economy: Empirical Research on its Application to Iranian State-Owned Enterprises. Long Range Planning, 49(5): 570-583. Azadegan, A., Srinivasan, R., Blome, C., & Tajeddini, K., . 2019. Learning from near-miss events: An organizational learning perspective on supply chain disruption response. International Journal of Production Economics, 216: 215-226. Tajeddini, K., & Mueller, S. L., . 2019. Moderating effect of environmental dynamism on the relationship between entrepreneurial orientation and firm performance Entrepreneurship Research Journal, 9(4): 1-13.

Author Response

Dear reviewer:

We are glad that you have carefully reviewed my paper and made valuable comments. We read your comments carefully and revised the paper. We explain your review comments as follows:

We have added some new references, including some you recommended.

And in this part, we have strengthened the discussion and explanation of regional innovation. We added a Roadmap(Fig. 1) .The references you recommended are excellent and helpful for us. We want to thank you again.

We supplemented the analytical steps of the theoretical model. We rewrote the contribution of the paper based on your recommendation. Strengthen the analysis of empirical results. It's mainly in “Estimated Results of the Spatial Econometric Model” and ” Regional Tests”.

We hope you satisfied with the revision of our article. If you think there are any shortcomings in our article, please point it out directly. We will revise it to your satisfaction.

Thank you again for your work.

Best wishes!

Authors of this paper.

Round 2

Reviewer 1 Report

Well done. Still need to proofread the paper. Congratulations!

Reviewer 2 Report

Thank you for improving your paper. I will be happy to recommend the paper for publication.